# The Environment Encouraging COVID-19 Response at Public Health Centers and Future Challenges in Japan

**DOI:** 10.3390/ijerph19063343

**Published:** 2022-03-11

**Authors:** Eri Osawa, Hiroko Okuda, Kyoko Koto-Shimada, Akira Shibanuma, Tomoya Saito

**Affiliations:** 1Department of International Health and Collaboration, National Institute of Public Health, Saitama 351-0197, Japan; 2Department of Health Crisis Management, National Institute of Public Health, Saitama 351-0197, Japan; okuda.h.aa@niph.go.jp; 3Department of Community and Global Health, Graduate School of Medicine, The University of Tokyo, Tokyo 113-0033, Japan; kkoto-shimada@umin.ac.jp (K.K.-S.); shibanuma@m.u-tokyo.ac.jp (A.S.); 4Center for Emergency Preparedness and Response, National Institute of Infectious Diseases, Tokyo 162-8640, Japan; saitot16@niid.go.jp

**Keywords:** COVID-19, public health center, public health activity, supportive environment, challenge, competency, Japan

## Abstract

The coronavirus disease 2019 (COVID-19) pandemic has triggered the implementation of public health measures globally. The health department of local governments has played a critical role in confronting COVID-19. In Japan, public health centers (PHCs) are focal points for COVID-19 response. Understanding the response to COVID-19 in local areas is critical to ensure adequate preparation for future emergencies. Therefore, the purpose of this study is to clarify how the COVID-19 operations by PHCs in Japan were managed and facilitated at the beginning of the infection spread, and their future challenges. We designed a case study that included two PHCs with a population of approximately 400,000 in Japan. Semi-structured focus group interviews with public health nurses from these two PHCs were conducted in September and October 2020. The data were analyzed using chronological time-series analysis. The switch to crisis response was encouraged by the business continuity plan. Their operations for the prevention of COVID-19 in the community were facilitated by the existing network. Further, strengthening the knowledge and skill regarding infectious disease control and management skills during infectious disease-related health emergencies were recommended. It is important to ensure that the environment facilitates emergency response and that people-and-community-centered health promotion activities are conducted, during an emergency situation, with more innovative action and leadership.

## 1. Introduction

Coronavirus disease 2019 (COVID-19), which was first identified in China in December 2019, rapidly spread to gain pandemic status [1]. In Japan, the first case of COVID-19 was confirmed on 15 January 2020 [2]. Subsequently, Japan has experienced a few waves of infection. Public health centers (PHCs), that are focal points for COVID-19 in communities, have maintained their operation for COVID-19 by changing their systems [3]. As of 2020, there are 469 public health centers established by prefectures and large cities nationwide [4]. The organizational structure of a PHC varies from one local government to another. In general, however, it consists of an infectious disease control section, a community health section (such as health promotion, mother and child health, mental health etc.) and a community welfare section, as health and human services, and a community hygiene section as environmental health services, with a director of the PHC, who is generally a doctor. Each public health center is staffed by doctors, pharmacists, public health nurses, dieticians, clinical laboratory technicians, and other specialists. The number of staff at each PHC varies, but according to national statistics, the number of full-time staff at public health centers nationwide was 57,207 in 2019 [5], indicating that, in recent times, the average number of staff is 122 per PHC. The duties of PHCs, as specified by the Community Health Act, are to act against infectious diseases as well as to conduct community health activities, such as maternal and child health, mental health, food sanitation, and environmental hygiene [6].

Normally, staff in PHCs contact residents through conducting health checkups, visiting the community to understand people’s living and health conditions, and providing continuous health guidance to solve their problems. Regarding the COVID-19 response, both prefectural and city PHCs perform the same tasks. Their operations focus on the prevention of infections through identification of the infected, contact tracing, cluster facility surveys, consultation, and coordinating hospitalization for the infected, but they do not include providing medical services [3,7]. Japan’s core strategy of COVID-19 response was a cluster-based approach, involving the investigation of retrospective clusters of COVID-19 cases. Retrospective cluster surveys with vigorous case investigations and subsequent COVID-19 testing revealed the source patients or clusters of over 60% of confirmed COVID-19 cases. This supported the implementation of the cluster-based approach at the community level [8]. In Japan, business operators are required to formulate a business continuity plan (BCP) as part of the new influenza action plan for an emergency situation of pandemic influenza and emerging infectious diseases under the Act on Special Measures for Pandemic Influenza and New Infectious Diseases Preparedness and Response [9] enacted in 2012 after the global experience of the H1N1 influenza pandemic of 2009. Local governments or/and PHCs also have been voluntarily developed BCP.

Health departments of counties across the US have also conducted contact tracing to reduce infection [10,11]. They faced difficulties due to people’s reactions, such as unresponsiveness to calls from local public health authorities, reluctance to disclose information because of lack of trust in the government, and limited operational resources of health departments [12]. It is indicated that local governments play a critical role in confronting COVID-19, and understanding the mechanism of the response to COVID-19 in local areas is necessary [13,14]. However, there are limited studies on COVID-19 response and the challenges faced by local governments in the front line. The purpose of this study is to identify how the COVID-19 response by PHCs in Japan was managed and facilitated at the onset of infection and the challenge to ensure future preparation. In particular, we will identify the major operations and the supportive environments for operations, as well as the future challenges, along with the stages of infection from the end of 2019, when the first case was reported globally, to October 2020.

## 2. Materials and Methods

### 2.1. Study Design

We employed a case study research method with semi-structured focus group interviews to understand the operations at PHCs and the facilitation of the operations for COVID-19 control at the frontline.

### 2.2. Study Setting

The study settings were two PHCs with a population of approximately 400,000 in each jurisdiction. PHC X was established by a prefecture, while PHC Y was established by a city. Both PHCs were located in suburban cities and not in Tokyo.

### 2.3. Study Participants and Data Collection

First, we selected two health centers using a non-random and purposive sampling method. In order to obtain the data, two focus group interviews with staff in each PHC were conducted by three members of the research group during September and October 2020. We chose public health nurses (PHNs) as interviewees because they represent about half of the PHC staff nationally, are full-time in every PHC, and play a critical role in their communities regarding the control of COVID-19 at PHCs [15]. Since we were focusing on supportive factors and future challenges rather than the COVID-19 operational response itself, we requested interviews with managerial- or leader-level PHNs who could grasp the entire workflow in a PHC. Finally, we interviewed seven PHNs. There were three and four participating PHNs from PHCs X and Y, respectively. The interviews were recorded with the consent of all participants, and verbatim transcripts were obtained. Along with focus group interviews, we collected documents with information on infection status up until the interview, organizational structure, and summary reports of operations. We explicitly enquired about their ever-changing operations and facilitating environment during different times, in chronological order, from the end of 2019, when the first case was reported globally, to the interview period (September and October in 2020).

### 2.4. Analysis of Data

The COVID-19 situation had changed sequentially, and related operations had also changed over time, and thus we employed chronological time-series analysis [15,16] to trace these changes. We analyzed the transcripts of the interviews in chronological order, grouped similar events, and conceptualized them based on our research questions: “What were the major responses to COVID-19?”, “How was the system changed in responding to COVID-19?” and “What facilitated or supported the system change?” For the chronological time-series analysis, we referred to the fourth pandemic phase, which was applied to preparedness planning for the H1N1 influenza pandemic [17] (Table 1). We extracted sentences related to challenges for the next pandemic from the transcripts and conceptualized them based on our question: “What are the challenges raised for the future, based on recent experiences?” To ensure the validity of the analysis, two researchers performed multiple and cross-analyses of the transcripts.

## 3. Results

### 3.1. Characteristics of the Public Participants

All seven of the PHN interviewees were female. They were managerial-level PHNs, such as the chief of the section for prevention of infectious diseases or community health and welfare, with over 20 years’ experience (from 20 to 36 years).

### 3.2. Responsible Operations for Controlling COVID-19 and System Changes at the PHC in Each Phase

Table 2 shows the outbreak stages divided into information from the currently collected data, responsible operations for controlling COVID-19 as major operations, system changes for the improvement of operations, and securing human resources and structural changes as supportive environments for operations. First, we collected data regarding the infection outbreak stage for H1N1 influenza (Table 1). Based on the data collected in interviews, time was divided into four periods, from the first COVID-19 case detected worldwide, and in Japan, until September 2020. We categorized January to February/March as the initial period because the first case in PHC X’s and PHC Y’s jurisdiction was found at the end of March and at the beginning of March, respectively—a month and a half to two months behind the first cases in Japan. Then, the second stage was the first wave period from March to April, being the time after the first case to the settling down of the first wave period of the infection. The third stage was the interlude period from May to June, followed by the fourth stage, being the second wave period from July to September.

#### 3.2.1. Initial Period (From January to February/March 2020)

Responsible operations and system changes in the initial period

During the initial period, neither PHC experienced positive cases. They started a telephone consultation and managed outpatient clinics for returnees from overseas and close contacts of infected individuals in accordance with the communication from the Ministry of Health, Labour and Welfare. At this stage, they had a regular meeting within the center and coordination meetings with stakeholders in the jurisdiction.

Structural changes in the initial period

To maintain responsible operations, they first changed staffing within the section. Their operations and system changes in the initial period were facilitated by the activation of the BCP by a prefecture or a city prepared for the case of an emergency.

#### 3.2.2. First Wave Period (From March to April 2020)

Responsible operations and system changes in the first wave period

After recognizing the first case in each PHC, they initiated contact tracing, cluster facility surveying, and coordinating hospitalization of infected individuals. They also provided home care support because some patients stayed at home. In particular, according to the interviews from PHC Y, they supported non-Japanese people because many foreigners resided in the area. They also prepared and managed data to check outbreak trends and were tasked with preparing information for press releases. To secure operations for COVID-19 in the first wave period, normal operations, such as group infant health check-ups and health promotion activities, were suspended. Workflows and manuals for operations were created based on their experience and the coordinated use of interpretation for the non-Japanese population.

Structural changes in the first wave period

In this stage, human resource support began to expand beyond either the section within the center or division (department) within the local government. They started employing temporary personnel to deal with their responsibilities. Increased staffing supported system changes to improve operations.

#### 3.2.3. Interlude Period (From May to June 2020)

Responsible operations and system changes in the interlude period

The first wave period settled in May 2020. During the interlude period, the PHCs, especially PHC X, employed surveys and workshops for the relevant facilities and organizations in their jurisdiction as preparation for the next outbreak. They checked the risks and preventive measures in facilities and disseminated information regarding the prevention of infection spread to relevant facilities and organizations. They also supported relevant organizations in creating a manual to deal with COVID-19 infections. They also reviewed their operations during the first wave period to prepare for the next outbreak. Parts of the normal operations, suspended in the first wave period, resumed.

Structural change in the interlude period

At this stage, there were no large structural changes or reinforcing personnel because it was not considered an emergency situation. According to the interview with PHNs at PHC X, activities for prevention in preparation for the next outbreak were possible because of the good relationships with relevant agencies and municipalities in normal times.

#### 3.2.4. Second Wave Period (From July to September 2020)

Responsible operations and system changes in the second wave period

A larger outbreak than the first wave of infections started in July for both PHCs. From July to September 2020, both PHCs expanded the capacity of the polymerase chain reaction test (PCR). In addition, they also made time to coordinate transportation, medical services, and hospitalization for the infected. Given the outbreaks among the same groups, cluster surveys were also part of the work during this period. Concurrently, the PHC provided information on prevention and infection control to facilities. In order to handle these urgent and complicated tasks, they revised forms and records in response to changes in circumstances and organized manuals and handbooks to support COVID-19 infected persons in their communities. Due to increased support from other sections and longer work hours, operations were performed in shifts, roles were reorganized, and orientation was provided from health-related staff to staff from other sections.

Structural change in the second wave period

In this stage, multisectoral support was provided within the local government and from external organizations, such as nursing associations, university faculties, and private human resource banks. PHC X started operations for monitoring the daily symptoms of the infected individuals in the private sector. These multisectoral and external supports maintained the operation of the PHCs.

### 3.3. Supportive Environment and Challenges Learned from COVID-19 Response from January to September 2020

Based on the interviews, the following were considered as supportive environments, challenges, and improvements for future action regarding the ongoing COVID-19 pandemic and emerging infectious diseases.

#### 3.3.1. Supportive Environment

Preparedness of the BCP encouraged emergency responses

The BCP encouraged them to switch from their usual work to emergency mode and started the COVID-19 response.


*“The director of the PHC instructed to work under the BCP.”*



*“After the BCP was put into operations, we reduced normal operations.”*



*“We did operations based on the H1N1 influenza action plan.”*


Dispersal allocation of PHNs to various departments

PHNs were able to exchange information on COVID-19 transmission and countermeasures with the relevant facilities quickly because they were assigned to responsible departments for these facilities. This made it easier to gather information.


*“There are also PHNs in the nursery section related to the nursery school and in the section of long-term care for the elderly. So, we always cooperate with these divisions.”*


Face-to-face relationships in normal time

A face-to-face relationship among other divisions and departments within the local government, stakeholders, and relevant organizations in the community, in normal times, encouraged information exchange and collaboration in response to COVID-19.


*“We had other infectious diseases occurring such as Norovirus and O157 in various places like schools and facilities in our jurisdiction before the COVID-19 outbreak. So, we always exchange in close communication with staff in relevant facilities.”*



*“We had a liaison group with infection control nurses in the jurisdiction for daily infection control in normal time before the COVID-19 outbreak. They helped us with training for facilities of the elderly and disabled.”*



*“Since we have been connected with the member in the Disaster Management Assistant Team (DMAT, which is supporting the COVID-19 control center) in our area before the COVID-19 outbreak, I think that our operations are functioning well.”*


Practical skills acquired from the former experiences

Practical skills acquired from previous work experiences, such as control of other infectious diseases and training for natural disaster response, were applied to the present outbreak.


*“PHNs who have experience with infectious diseases response can take smart action. They quickly and appropriately assess the situation just by taking a look at the summary of consultations with residents.”*



*“At the PHC, we had a DMAT drill with over 100 teams. This kind of training was also very useful for us this time. For example, we utilized skills such as writing the chronology of a situation, making to-do lists, and assigning roles.”*


#### 3.3.2. Challenges and Improvements for Future Action

Staff knowledge of specialized infection control

It was necessary for more staff members to undergo training related to infectious disease control. What was also needed in this regard was a reinforced job rotation system that allowed staff to experience the work of various departments, including the infection control department. Staff felt the need to be prepared to explain and teach the work to those who had no experience in health risk management.


*“I think that experiencing infectious disease control at least once by job rotation is important.”*



*“I think the most realistic approach for us would be to have the number of Field Epidemiology Training Program participants increase within the prefecture, and then the PHC would receive support to improve our knowledge and skills from them.”*


Management skill

Other competencies that needed improvement were management skills, such as understanding the situation comprehensively and predicting future occurrences.


*“It is necessary to be able to look at any situation from a bird’s eye view, and to be able to see what is going on now and what will happen in future.”*


Accepting diverse cultures

Based on experience in supporting non-Japanese people in the COVID-19 response, it was pointed out that competency in accepting diverse cultures was also necessary.


*“There were many non-Japanese people who tested positive; hence, we had difficulties in communicating with them because of the language barrier, different cultures, and different lifestyles.”*



*“I think we need to hone our skills a little more in terms of globalization and acceptance of diversity.”*


Digitization of information

The digitization of information must be improved and will aid in more efficient future operations.


*“I think that we can still improve the way we digitalize information and use the various technological devices of today. There is room for change through education.”*


## 4. Discussion

In this study, we chronologically analyzed PHCs’ operations regarding the COVID-19 response and its encouraging background and challenges for the future in Japan. We will discuss three points of our study results: Using the BCP as a switch to emergency response, collaboration with the community as supportive environments for activities in emergency situations, and competencies needed to respond to emerging infectious diseases in the future.

### 4.1. Using the BCP as a Switch to Emergency Response

There are necessary tasks that must be conducted according to the Infectious Disease Control Act, as COVID-19 is a designated infectious disease under tight control. However, the BCP that was prepared seemed to be useful for switching from normal operations to crisis management operations in response to COVID-19. The study showed that the BCP was helpful for organizations after the earthquake in 2010 and 2011 to maintain business in New Zealand [18]. BCPs, assuming region-specific disasters, are also applicable to various types of disasters [19]. There are reports from Japan that the BCP failed during the present COVID-19 outbreak [20]. It will be necessary to create a concrete and realistic BCP at a level where action can be taken in an emergency. The effectiveness of BCP in unexpected situations and long-lasting outbreaks of infectious diseases, such as COVID-19, needs to be verified.

### 4.2. Collaboration with the Community as Supportive Environments for Activities in Emergency Situations

In the interlude period, the preventive intervention was conducted by the PHC for relevant facilities in their jurisdiction. Direct information dissemination and health education to relevant agencies are recognized as health promotion and community engagement strategies that are vital for critical functions during emergencies [21,22]. The success of risk communication and community engagement programs rely on strong associations and engagement among various partners in a particular area [23].

PHC Y’s health promotion and community engagement strategy may be attributed to the fact that they considered the COVID-19 response as a part of their activities for community-based integrated care systems that have been progressively built in collaboration with other agencies in the community.

### 4.3. Competencies Needed to Respond to Emerging Infectious Disease Outbreaks in the Future

Regarding future challenges, we found some aspects of competencies to respond to emerging infectious disease outbreaks. One of these was knowledge and practice in dealing with infectious diseases, and the other was management skills under emergency conditions. The results showed that disaster response training before the current pandemic was useful in managing the COVID-19 response. There is a strong possibility that natural disasters will occur during an infectious disease pandemic. Therefore, integrated training for the management of infectious disease response with disaster response is suggested. Through the COVID-19 pandemic, the need for blended skills of scientific research and analysis, as well as sensitivity to social inequalities with ethics and respect for human diversity, have increased [24]. The fact that PHC X experienced difficulty in dealing with non-Japanese people in the current study also supports this. It has been reported that public health leaders demonstrating critical thinking, system thinking, and transformative action, was reinforced during the early stages of the outbreak in Canada [25]. The experience of dealing with the COVID-19 pandemic will add more innovative competency. Further studies on human resource development for infectious disease-related public health emergencies are needed.

### 4.4. Limitations of Study

This study had some limitations. The participants were limited to only two centers in Japan. Generalization to all public health agencies might be inappropriate because the COVID-19 response is largely affected by regional characteristics and the subjective situation of infected areas. We asked the participants to remember their experiences from about eight months previously at most. Therefore, there may be an information bias. However, in addition to focus group interviews, reports about organizational structure and operations by PHNs were used to increase reliability.

## 5. Conclusions

This study preliminarily identified COVID-19-related operations by PHCs, how they managed and encouraged the frontline response in Japan, and the challenges they faced based on their experiences. During the current unexpected COVID-19-related health crisis, PHCs coped with the ever-changing situation by utilizing the prepared plan and existing network with relevant divisions and organizations, and reviewing their daily operations. They also implemented a people-and-community-centered approach, that is important for public health. To ensure an adequate level of future preparation, more innovative thinking and action than before, as well as competencies for infectious disease-specific response and management and leadership skills during health emergencies, are required. Countries worldwide should be encouraged to become involved in systems development and human resource development to enable their own local governments to implement feasible COVID-19 measures compatible with local characteristics.

## Figures and Tables

**Table 1 ijerph-19-03343-t001:** Infection outbreak stage for H1N1 Influenza indicated in the Japanese government action plan [17].

Stage	Explanation
1. Overseas outbreak	Outbreak of new infections overseas
2. Early period of domestic outbreak	(At the national level) Outbreak of new infections in any of the prefectures. However, all contacts can be traced by epidemiological survey. (At prefectural level) i. No outbreak ii. Early stage of outbreak/all contacts can be traced by epidemiological survey
3. Pandemic within a country	(At the national level) Stage when all contacts of new case cannot be traced in any of the prefectures by epidemiological survey.
4. Period of Interlude	Remaining low infection

**Table 2 ijerph-19-03343-t002:** Responsible operations for controlling COVID-19 and supportive environment for operations from January to September 2020.

	Initial Period	First Wave Period	Interlude Period	Second Wave Period
Infection outbreak stage corresponding to Avian Influenza action plan (Table 1)	1. Overseas outbreak 2-i Early stage of domestic outbreak (No cases in a prefecture [PHC’s jurisdiction])	2-ii Early period of outbreak (Early stage of outbreak in a prefecture [PHC’s jurisdiction])	4. Period of Interlude	3. Pandemic within a country (Pandemic within a prefecture [PHC’s jurisdiction])
Time	From January to February/March 2020	From March to April 2020	From May to June 2020	From July to September 2020
Responsible operations for controlling COVID-19	Starting telephone consultation Managing outpatient clinics for returnees from overseas and their close contacts	Contact tracing Cluster facility survey Coordination of hospitalization Monitoring daily health status of the infected Home care support Foreigner support Data preparation and management Preparation for press release	Checking risks and preventive measures for relevant facilities Having workshops for information dissemination Support for developing manuals for relevant organizations Having review session of operations in the first wave period	Expanding PCR testing Contact tracing Cluster facility surveys Monitoring daily health status of the infected Coordinating medical services, transportation for infected patients Information dissemination to facilities (for the elderly and the disabled)
System changes for improvement of operations	Having regular meetings Having coordination meeting with stakeholders	Suspending normal operations (such as group infant health check-ups, health promotion activities for member of communities) Creating workflows and manuals for operations Coordinating use of interpreters and function of interpretation for non-Japanese population	Resuming a part of normal operations (such as group infant health check-ups, health promotion activities for member of communities)	Revising forms, records, etc. in response to changes in circumstances Organizing manuals and handbooks for supporting COVID-19 infected person in communities Setting work hours and work shifts. On-the-job-training (OJT) from health-related personnel to other personnel Re-clarifying roles of operations
Securing human resources and structural change	Activating Business Continuity Plan Support within the section and partially from other section	Support by administrative officers across other sections Centralized COVID-19 consultation center was controlled by Disaster Management Assistant Team (DMAT) Temporary personnel (especially for Public Health Nurses)		Multi-sectional support Support from external experts for cluster survey team External professional support (prefectural nursing associations, university faculty, private human resource banks) Business outsourcing (private sector utilization)

## Data Availability

The data presented in this study are not publicly available due to privacy or ethical restrictions.

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
