# Peer review of "The Environment Encouraging COVID-19 Response at Public Health Centers and Future Challenges in Japan"

_ijerph, 2022, doi:10.3390/ijerph19063343_

Round 1

Reviewer 1 Report

  1. Introduction:
  • For understanding of reader, describe the structure of PHCs, including total staff in each center, and different categories. How PHC staff approach the community and maintained linkages.
  • In Japan cluster-based approach strategy was implemented to investigate retrospective cluster of COVID-19 cases. Highlight the importance of strategy and acknowledge the research work done by other researcher.

  1. Methods:

Study setting:

  • Describe how two PHCs were selected? (Random or non-random selection)?

Study participants and data collection:

-             Describe how focus group interviews/discussions were conducted and who conducted these interviews?

-             What was criteria for selection of participants (public health nurses)? Why only public heath nurses were selected as study participants?

-             Along with focus group interviews other information was also collected. Describe briefly how and what additional information was collected?

  1. Results
  • Link the results with clearly defined objectives of the study (How the COVID-19 response by PHCs was managed and facilitated).
  1. Discussion:
  • Discuss the important findings of the study.
  • Discuss the interpretation of results with reference to work done on the topic by other researchers.
  1. Conclusion
  • The conclusion should help the reader to understand why your research matters to them.

Reviewer 2 Report

The authors examined the use of public health tools to deal with the Covid-19 pandemic in Japan on the example of selected two public health centers covering a population of approximately 400,000 people.

The article addresses a very important issue from the point of view of treatment management during a pandemic around the world.

I have the impression that the submitted article has the character of a report rather than a scientific article, despite the fact that it contains an analytical part. The interviews were conducted and the way in which they were prepared is very good.

An additional advantage of the article is the issue of the importance of local governments in the organization of disease management. Again, this is a challenge in many other countries, especially those with a tendency towards centralized management. Therefore, this article provides an important platform for the exchange of experiences and views on the use of public health - especially at the level of local governments.

Point 2 / shows that the study was carefully planned and carried out. In point 3 / the results were clearly presented and commented on. In point 4 / some international comparisons were made, and this always raises some doubts due to the strong differentiation of conditions, which remains without a proper comment. The essence of the article was included in the conclusions.

The article is clearly written taking into account the individual steps of the research. It includes a chronological and thorough cause-and-effect analysis. It is of interest to health policymakers in other countries and, of course, to health researchers.

Of course, the authors analyzed disease management under specific Japanese conditions. However, it is a platform for drawing conclusions for the needs of other countries with completely different conditions. It may provide a kind of pattern for considerations in other countries with a differently organized health care system and a different role for public health.

Reviewer 3 Report

The authors present qualitative research to describe the efforts undertaken in public health centers in Japan by interviewing seven public health nurses across two centers. The interviews yielded descriptions of changes to protocol taken during different periods during the first several months of the COVID-19 pandemic. Such information is helpful for understanding how the COVID-19 pandemic has been handled in Japanese health settings.

MAJOR COMMENTS

-The setup of the focus groups and interviews needs clarification. The authors describe focus groups, and then describe interviews occurring over a two-month period. Focus groups, by definition, require multiple interviewees being interviewed at the same time. If the four and three nurses at each PHC were interviewed together and represent two focus groups, this needs to be stated. If the seven nurses were interviewed separately without interviews of multiple participants, then focus groups were not conducted and should not be described.

-The data analysis needs clarification. Was chronological order in reference to when the interviews took place or were the interviewees asked explicitly about events that occurred during different time periods?

MINOR COMMENTS

-Lines 105-107: The text refers to the fifth pandemic phase from Table 1, but the table only lists four pandemic phases.

-Table 2: During the Interlude period, resuming a part of normal activities is mentioned. Are there any specific activities that were resumed?

Round 2

Reviewer 3 Report

Previous comments have been addressed. There is now more clarity in how the interviews were conducted along with a clearer summary of the interviews' contents.